# Engineered Human Dendritic Cell Exosomes as Effective Delivery System for Immune Modulation

**DOI:** 10.3390/ijms241411306

**Published:** 2023-07-11

**Authors:** Ranya Elsayed, Mahmoud Elashiry, Cathy Tran, Tigerwin Yang, Angelica Carroll, Yutao Liu, Mark Hamrick, Christopher W. Cutler

**Affiliations:** 1Department of Periodontics, Dental College of Georgia, Augusta University, Augusta, GA 30912, USA; 2Department of Cellular Biology and Anatomy, Medical College of Georgia, Augusta University, Augusta, GA 30912, USA

**Keywords:** dendritic cells, periodontitis, exosomes, immunotherapy

## Abstract

Exosomes (exos) contain molecular cargo of therapeutic and diagnostic value for cancers and other inflammatory diseases, but their therapeutic potential for periodontitis (PD) remains unclear. Dendritic cells (DCs) are the directors of immune response and have been extensively used in immune therapy. We previously reported in a mouse model of PD that custom murine DC-derived exo subtypes could reprogram the immune response toward a bone-sparing or bone-loss phenotype, depending on immune profile. Further advancement of this technology requires the testing of human DC-based exos with human target cells. Our main objective in this study is to test the hypothesis that human monocyte-derived dendritic cell (MoDC)-derived exos constitute a well-tolerated and effective immune therapeutic approach to modulate human target DC and T cell immune responses in vitro. MoDC subtypes were generated with TGFb/IL-10 (regulatory (reg) MoDCs, CD86^low^HLA-DR^low^PDL1^high^), *E. coli* LPS (stimulatory (stim) MoDCs, CD86^high^HLA-DR^high^PDL1^low^) and buffer (immature (i) MoDCs, CD86^low^HLA-DR^med^PDL1^low^). Exosomes were isolated from different MoDC subtypes and characterized. Once released from the secreting cell into the surrounding environment, exosomes protect their prepackaged molecular cargo and deliver it to bystander cells. This modulates the functions of these cells, depending on the cargo content. RegMoDCexos were internalized by recipient MoDCs and induced upregulation of PDL1 and downregulation of costimulatory molecules CD86, HLADR, and CD80, while stimMoDCexos had the opposite influence. RegMoDCexos induced CD25+Foxp3+ Tregs, which expressed CTLA4 and PD1 but not IL-17A. In contrast, T cells treated with stimMoDCexos induced IL-17A+ Th17 T cells, which were negative for immunoregulatory CTLA4 and PD1. T cells and DCs treated with iMoDCexos were immune ‘neutral’, equivalent to controls. In conclusion, human DC exos present an effective delivery system to modulate human DC and T cell immune responses in vitro. Thus, MoDC exos may present a viable immunotherapeutic agent for modulating immune response in the gingival tissue to inhibit bone loss in periodontal disease.

## 1. Introduction

Periodontitis (PD) is one of the most common inflammatory bone degenerative diseases in the U.S. [1] contributing to risks of other diseases with more serious mortality and morbidity profiles, particularly those in advanced age [2]. Currently lacking is an effective immunotherapy for PD. Investigations of the histopathology of PD lesions [3,4,5] in humans [6,7,8,9,10] and mice [11] support a pathological role for tissue resident matured dentritic cells (DCs)and T helper 17 (Th17^+^ CD4^+^ T) cells in alveolar bone loss. Recent studies have indicated that PD lesions are an organized cellular response, with tissue resident DCs, B cells, plasma cells, M1 macrophages [12,13], and CD4+ T cells [8]. Subgingival oral microbes, such as *P*. *gingivalis* [2,10] and Fusobacterium nucleatum [14], invade immature DCs and other cells in PD lesions. These species regulate DC maturation and inhibit signals for DC migration to draining lymph nodes and the initiation of immunity [2,15,16,17]. Persistent local inflammatory signals promote unrestrained activation of DC maturation in peripheral tissues in situ, as in Crohn’s disease [18], arthritis [19], and PD [6,7,8,9]. Intensive immune clusters of matured CD83+ DCs with CD4+ T cells are found in the lamina propria of PD patients, [6,7,8,9] evocative of ectopic lymphoid foci [20,21]. DCs can shape the Th17:T regulatory cell (Treg) balance toward Th17 to eradicate invading bacteria, but Th17:Treg imbalance promotes osteoclast-mediated alveolar bone loss [3,22]. Immune-regulatory DCs can also induce FoxP3+ Tregs, which have inhibited Th17 responses and attenuated inflamtory bone loss in PD animal models [4,5].

We propose here a novel therapeutic strategy for PD. Autologous exosomes (exos) derived from dendritic cells (DCs), the “directors” of immune response, are used as a natural, nano-sized delivery system for immunoregulatory cargo. Exos are nano-sized (30–150 nm) membrane-enclosed particles that originate in the endocytic pathway and are secreted by most cell types. Once released from a secreting cell, exos interact with the neighboring environment and deliver proteins, nucleic acids, lipids, microRNAs, and other signaling molecules to recipient cells, influencing their function and biological responses [23,24,25]. The therapeutic potential of exos is supported by favorable safety and tolerance profiles in phase-I/II clinical trials for cancers [26,27,28,29] and autoimmune diseases. Exosomics is a new approach to identify early cancer markers in saliva, with reduced contamination and data variability [30]. Promising preclinical data in mice showed gingival retention of injected DC exos, reprogramming of recipient DCs and T cells in gingiva, and differential regulation of alveolar bone loss, depending on exo subtype [11]. Further advancement to translate this technology into the clinical arena requires the testing of human DC-based exos with human target cells. This study investigates the immune functions of three human monocyte-derived dendritic cell (MoDC) exo subtypes, including regulatory MoDC exos (regMoDCexos), stimulatory MoDC exos (stimMoDCexos), and immature MoDC exos (iMoDCexos). RegMoDCexos are engineered to express immune-regulatory cargo by adding the anti-inflammatory cytokines TGFB1 and IL10 during the differentiation of the parent cells, or MoDCs, followed by additional active loading of the purified exos with TGFB1 and IL10 using ultrasonication. Our main focus is to test the ability of human regMoDCexos in promoting bone-protective CD25+Foxp3+T cell (Treg) response while inhibiting bone-resorbing Th17 cell induction in vitro [11]. This study highlights the efficacy of the local delivery of human regMoDCexos as a potential therapeutic modality for PD.

## 2. Results

### 2.1. Human MoDC Phenotypes and Cytokine Profiles Consistent with Murine DC Subtypes

Differentiation of human CD14+ monocytes into MoDCs was assessed through flow cytometry [9,10,31]. We showed a 91.8% population of bona fide human immature MoDCs (CD1c^+^CD14^−^ iMoDCs) (Figure 1A and Appendix A). From these MoDCs, subtypes were generated with TGFb/IL-10 (regMoDCs, CD86^low^HLA-DR^low^PDL1^high^), *E*. *coli* LPS (stimMoDCs, CD86^high^HLA-DR^high^PDL1^low^), and buffer (iMoDCs, CD86^low^HLA-DR^med^PDL1^low^) (Figure 1B–E). The cytokine mRNA profiles of MoDC subtypes were as follows: regMoDCs (very low IL-6/IL-1b/TNFa, very high TGFb/IL-10), stimDCs (high IL-6/IL-1-b/TNFa, low TGFb/IL-10), and iDCs (low IL-6/IL-1b,/TNFa, high TGFb/IL-10) (Figure 1F).

### 2.2. Characterization of Human MoDC-Derived Exos

Exos were purified from MoDC subtypes (Figure 2), quantitated, and characterized for correct size (~150 nm) distribution (Figure 2A) and cup shape (Figure 2C). A Western blot analysis (Figure 2B) revealed shared exo proteins CD81, TSG101, and CD9 and distinct cytokines in regMoDCexos (positive for TGFB-1, negative for IL-6, IL-1b, and TNFa), iMoDCexos (all negative), and stimMoDCexos (negative for TGFB-1, positive for IL-6, IL-1b, and TNFa). Immunogold TEM confirmed luminal and surface localization of TGFB-1 (Figure 2D) and IL-10 (Figure 2E), and CD63 surface expression (Figure 2F) in regMoDCexos.

### 2.3. MoDCs Take Up Exos, Altering Cytokine mRNA Profiles

MoDCs cocultured with Dil (red)-labeled MoDC exos were analyzed through confocal microscopy showing internalized exos (red) inside recipient MoDCs (green) (Figure 3A). Immunogold TEM demonstrated the colocalization of TGFB1 in MoDCs only when treated with regMoDCexos (Figure 3B). Cytokine profiling indicated that regMoDCexo uptake increased *IL-10* mRNA and decreased *IL-6*, *TNFa,* and *IL23* mRNA. In contrast, stimMoDCexos decreased *IL-10* and increased IL-6, *TNFa,* and *IL23* mRNA. The cytokine profile of iMoDCexos was the same as control exos (Figure 3C).

### 2.4. MoDC Maturation State Modulated by MoDC Exos

Recipient iMoDCs were cocultured with MoDC exo subtypes or no exos (Ctl) for 3 days and then analyzed using flow cytometry for the expressions (MFIs) of PDL1, CD86, HLADR, and CD80. RegMoDCexos induced upregulation of PDL1 and downregulation of costimulatory molecules CD86, HLADR, and CD80 on recipient MoDCs, while stimMoDCexos had the opposite influence on recipient MoDCs, stimulating DC maturation and inhibiting regulatory PDL1 (Figure 4). iMoDCexos were equivalent to control (no exos).

### 2.5. RegMoDCexos Modulate T Cell Effector Responses

Dil-labeled regMoDCexos were cocultured with CD4 T cells and were shown to colocalize with the recipient cells using immunolabeling and confocal microscopy imaging (Figure 5A). Human naïve autologous CD4+ T cells +/− anti-CD3/CD28 were cocultured with MoDC exo subtypes. RegMoDCexos induced 46% CD25+Foxp3+ Tregs (Figure 5B), which expressed CTLA4 and PD1 (Figure 5C,D) but not IL-17A (Figure 5E). In contrast, CD4 T cells treated with stimMoDCexos induced 26% IL-17A+ Th17 T cells, which were negative for immunoregulatory CTLA4 and PD1. T cells treated with iMoDCexos were immune ‘neutral’, equivalent to controls.

## 3. Discussion

In this study, we propose a novel therapeutic strategy for PD using nanoparticles called exosomes (exos) from human dendritic cells (DCs), the “directors” of immune response (Figure 6). Our immunotherapeutic strategy for PD (Figure 6), consisting of regulating local, mature DC and Th17 T cell immune responses, is well-grounded in the autoimmune/inflammation literature [18,19]. Moreover, our previously published results [11] support its efficacy in vivo in mice, although not in human studies. Hence, more translational studies on the development of human DC-based exos and their interactions with human recipient cells relevant to PD pathogenesis are deemed necessary. This is a fundamental step toward a potential cell-free immunotherapy in humans. Exos are nanoparticles of endosomal origin secreted by all cells, including DCs [32], and are found in all body fluids [33]. Proteomic and transcriptomic analyses of salivary exos show promise as early cancer markers [30]. PD patient saliva contains cytokines [34], as well as DC-exo-related tetraspanins CD9 and CD81 [35]. Exos are endogenous or eobiotic, negating off-target effects or foreign-body reactions [36]. Exos can deliver molecular cargo to promote cytotoxic T cell responses for cancer [26,27,28,29] or inhibit T cell responses in autoimmune diseases [37]. Exos have ideal traits for drug delivery, including small size (30–150 nm), long circulation time, low clearance, and cargo preservation [11,38]. We previously reported in a mouse model of PD that custom murine DC-derived exo subtypes could reprogram the immune response toward Treg bone-sparing or Th17 bone-loss phenotypes, depending on the immune profile [11].

Three different human MoDC subtypes were developed and utilized as the source of the tested MoDC exo subsets in the current in vitro study (Figure 1). Here, we showed the feasibility of generating human MoDC exos (Figure 2) with immune profiles corresponding to our reported murine DC exo counterparts. RegMoDCexo uptake by acceptor human DCs reduced DC maturation/activation and upregulated the inhibitory molecule PDL1, while the opposite stimulatory response was seen with stimMoDC exo treatment, suggesting the modulatory role of different human MoDC exo subtypes (Figure 3 and Figure 4). Moreover, regMoDCexo interaction with CD4 T cells promoted Treg response and inhibitory molecules’ CTLA4 and PD1 expressions while inhibiting Th17 cell induction. In contrast, stimMoDCexos enhanced the Th17/Treg ratio (Figure 5). This is in line with the findings of our published in vivo work in PD murine models [11] and with other animal studies performed by different groups [4,5].

Although DC exos are eobiotic and well-retained at the gingival site of injection, we recognize that trace levels of proteins and nucleic acids (as we reported in [11]) in DC exo subtypes could induce unforeseen, off-target effects locally or at distant tissue/organ sites. Thus, we monitored the effects on the liver and kidneys in mice where a blinded, pathologist (Z.K.) assessed liver and kidney sections for histopathologic evidence of toxicity (Appendix A), as reported [39]. No evidence of cytotoxicity was observed with exo treatment in the liver or kidneys, and no changes in immune cell populations were evident in the spleens of treated mice (Appendix A). However, pharmacokinetics/pharmacodynamics (PK/PD) and safety profiles should be confirmed in large animals, such as nonhuman primates.

TGFß [40] and IL-10 [41] are particularly appealing as immune-regulatory cytokines for loading into DC exos. Our data showed that TGFB-1 and IL-10 were localized on the outer surface, as well as inside regMoDCexos, as shown through TEM (Figure 2D,E). In addition, TGFB-1 was shown to colocalize with regMoDCexo-treated MoDCs, suggesting the delivery of the immunoregulatory TGFB1 cargo to the target cells (Figure 3B). TGFB-1 is pleiotropic, with high TGFB-1 levels activating SMAD2/3, inhibiting DC maturation, suppressing Th1 and Th2 T cells via T-bet and GATA-3, and promoting bone-sparing FoxP3+ Tregs [40]. On the contrary, low TGFß levels and high IL-6 can promote Th17 cell differentiation through RORýt [42]. Disruption of the SMAD pathway delays periodontal repair [43]. A wealth of data support the IL-10/STAT3 axis as a major transcriptional inhibitor of immune response genes to inhibit DC maturation and promote Tregs [41]. Further studies are required to examine the specific roles of these molecules in target phenotypes in vitro. Collectively, the results from this study strongly suggest that MoDC exos may be a viable immunotherapeutic agent for modulating human DCs and T cells to inhibit bone loss in PD.

## 4. Materials and Methods

### 4.1. Generation and Cultureing of Human Monocyte-Derived DCs

Buffy coats were isolated from peripheral blood through density gradient centrifugation on a Lymphoprep instrument (STEMCELL Technologies, Vancouver, BC, Canada; Cat# 07801). Approximately 10^9^ peripheral blood mononuclear cells (PBMCs) from the buffy coats were collected, washed, and stored at −80 °C until further processing. Approximately 10^8^ human CD14+ monocytes were isolated from PBMCs with a negative selection isolation kit (EasySep, STEMCELL Technologies, Vancouver, BC, Canada; Cat#19359). Cells were cultured at a density of 10^6^ per ml in exo-depleted X-Vivo 15 growth medium supplemented with GMCSF (100 ng/mL) (Peprotech, Rocky Hill, NJ, USA; Cat#315-03) and IL-4 (50 ng/mL) (Peprotech, Rocky Hill, NJ, USA; Cat# 214-14). Medium was changed every 2 days, and cells were harvested on day 6 to generate iMoDCs. To generate stimMoDCs, part of the harvested cells were treated with 100 ng/mL *E. coli* LPS (Sigma, St. Louis, MO, USA) on day 5 of culturing, and cells were harvested on day 6. To generate regMoDCs, 50 ng/mL TGFb (R&D Systems, Inc. Minneapolis, MN, USA) and 70 ng/mL IL10 (R&D Systems, Inc. Minneapolis, MN, USA) were added to fresh medium on day 4, and cells were harvested on day 6. Differentiation of human CD14+ monocytes into different MoDC subtypes was assessed via flow cytometry and qPCR.

### 4.2. Exosome Isolation and Engineering

Exosome isolation was performed as previously described [23,24] (Elsayed et al. 2022, Elsayed et al. 2021). Briefly, in order to remove debris and cells, the supernatants from MoDC cultures underwent successive centrifugations at 500× *g* for 5 min, 2000× *g* for 20 min, and 10,000× *g* for 30 min. Afterward, ultrafiltration 2 times with 0.2 u µm and 2 times with 100 kDa filters (to remove microvesicles and free proteins) and ultracentrifugation for 1.5 h at 120,000× *g* were performed. The exosome pellets were then washed with PBS, ultracentrifuged 2 times for 90 min at 120,000× *g*, subsequently resuspended in 100 μL PBS, and stored at –80 for further studies. For engineering of regMODCexos, 1 × 10^9^ particles was actively loaded through sonication with 5 ug TGFB1 and 5 ug IL10 in 500 µL PBS, filtered 3× through ultrafiltration with a 100 KDA filter to remove free proteins, washed 3× with PBS, ultracentrifugated at 120,000× *g* for 1.5 h to further purify exos from free molecules, and finally resuspended in 100 µL PBS. The supernatant in which regMoDCexos were suspended was then isolated and analyzed for any contaminants via ELISA. The yields of the isolated exosomes from their respective MoDC subtypes were as follows:RegMoDCexos: approximately five vesicles per one cell per hour.IMoDCexos: approximately five vesicles per one cell per hour.StimMoDCexos: approximately nine vesicles per one cell per hour.

### 4.3. Immunogold Plating and Transmission Electron Microscopy (TEM)

As described previously [12,13], exosome samples were fixed overnight in 4% paraformaldehyde in 0.1M cacodylate buffer at PH 7.4. On a carbon–Formvar-coated 200 mesh nickel grid, five microliters (5 µL) of suspended exosome preparation was applied and allowed to stand 30 min. Whatman filter paper was used to wick off the excess sample. With the exosome side down, grids were floated on a 20 µL drop of 1M ammonium chloride for 30 min to quench aldehyde groups from fixation steps. Afterward, grids were floated on drops of blocking buffer (0.4% BSA in PBS) for 2 h and then rinsed 3 times for 5 min each with PBS. Grids were set up as follows and were allowed to incubate in blocking buffer or primary antibody (anti-TGFb, anti-IL10, or anti-CD63) for 1 h. For 1 h, grids were floated on drops of 1.4 nm nanogold secondary antibody (Nanoprobes, Inc., Yaphank. NY, USA), which was diluted at 1:1000 in blocking buffer. Grids were then rinsed 3 times for 5 min each with DI H2O. Exosome samples were postfixed in 2% osmium tetroxide in NaCac buffer and dehydrated in ethanol for visibility in electron SEM. Then, samples were mounted on aluminum stubs and sputter-coated for 6 min with gold–palladium (Anatech Hummer 6.2, Union City, CA, USA). Using an FEI XL30 scanning electron microscope (FEI, Hillboro, OR, USA), exosomes were observed and imaged at 10 kV.

### 4.4. Nanotracking Analysis

A nanoparticle-tracking analysis (NTA) with a Zeta View PMX 110 instrument (Particle Metrix, Meerbusch, Germany) was used for visualization and analysis of size and count of nanoparticles in suspension. Briefly, as previously described [11,23,24] (Elsayed et al. 2022; Elsayed et al. 2021; Elashiry et al. 2021), data about size and concentration were acquired with ZetaView software (8.02.28) after loading 10 μL of exosome sample into the sample chamber.

### 4.5. Western Blot

As previously described by our group [23], exosomes were lysed with the addition of RIPA buffer with a protease/phosphatase inhibitor cocktail and incubated for 20 min on ice. Denatured proteins were separated using 4–15% Mini-PROTEAN TGX Precast Protein Gel (Bio-Rad labratories, Hercules, Ca; Cat#4568084) and then transferred onto PVDF membranes (Bio-Rad labratories, Hercules, Ca; Cat#1620177). After blocking with 5% nonfat dry milk in TBST for 1 h, membranes were incubated with primary antibodies at 4° overnight. The membranes were washed with TBST, and then incubated with HRP-conjugated secondary antibodies for 1 h at room temperature. Membranes were washed and developed with an ECL kit (Thermofisher Scientific, Waltham MA, USA; Cat# 50-210-368) and imaged with ChemiDoc MP Imaging Gel (Bio-Rad labratories, Hercules, CA, USA). Antibodies used were anti-CD81 (#10037) (Cell Signaling Technology, Danvers, MA, USA), anti-TSG101 (MA1-23296) (Thermofisher Scientific, Waltham MA, USA), anti-CD9 (#13174) (Cell Signaling Technology, Danvers, MA, USA), anti-TGFb (#3709) (Cell Signaling Technology, Danvers, MA, USA), anti-IL-6 (#12153) (Cell Signaling Technology, Danvers, MA, USA), anti-IL-1B (#12703) (Cell Signaling Technology, Danvers, MA, USA), and anti-TNFa (#6945) (Cell Signaling Technology, Danvers, MA, USA). Secondary antibodies used were anti-mouse IgG HRP-linked (#7076) (Cell Signaling Technology, Danvers, MA, USA) and anti-rabbit IgG HRP-linked (#7074) (Cell Signaling Technology, Danvers, MA, USA) antibodies.

### 4.6. qPCR

Total RNA was isolated from DCs using a QIAGEN RNeasy mini kit (Qiagen, Inc., Valencia, CA, USA; Cat#: 74134). A Nanodrop instrument (NanoDrop 1000 UV-VIS Spectrophotometer Software Ver.3.8.1, Thermofisher Scientific) was used to assess RNA concentration and purity. A ratio of 260/280 of 2.0 was considered suitable for further analysis. A High-Capacity Reverse Transcription Kit (Applied Biosystem, Thermofisher Scientific, Waltham MA, USA) was used to perform reverse transcription to cDNA in a total reaction volume of 20 µL. Quantitative real-time PCR was performed using TaqMan fast advanced master mix (Applied Biosystem, Thermofisher Scientific, Waltham, MA, USA) and TaqMan Gene Expression assays (Applied Biosystem, Foster City, CA, USA) specifically for IL6 (Hs00174131_m1), IL1 (Hs000961622_m1), IL-1B (Hs01555410_m1), TNFa (Hs01113624_g1), IL23 (Hs0372324_m1), and internal control GADPH (Hs02758991_g1). RT-PCR was run in a StepOnePlus Real-Time PCR System. Calculation of relative gene expression was performed using delta-delta CT and plotted as relative fold change.

### 4.7. Flow Cytometry and Antibodies

Staining of cell suspensions was performed on ice with FACS Staining Buffer (Thermofisher Scientific, Waltham, MA, USA). Blocking of FC receptors (FCRs) was performed using human FCR-blocking reagent (Miltenyi Biotec) for 15 min on ice and protected from light. Fluorophore-conjugated antibodies were added at the recommended concentration on ice for 30 min. Cells were then washed and resuspended in FACS buffer. After fixation and permeabilization using a fixation/permeabilization buffer set (eBioscience, Thermofisher Scientific, Waltham, MA, USA), intracellular staining was performed according to the manufacturer’s protocol. Conjugated antibodies in 1× permeabilization buffer were added, and cells were incubated while covered from light for 1 h. Cells were washed and resuspended in FACS buffer. Data were acquired and analyzed using a MACSQuant flow cytometer machine and corresponding software MACQuantify 2.13.3 (Miltenyi Biotech, Auburn, CA, USA). Antibodies used were anti-human CD1c APC, clone: L161 (Affymetrix, eBioscience, Thermofisher Scientific, Waltham MA, USA; cat#: 17-0015-41); anti-human HLADR Brilliant Violet 421, clone: G46-6 (Affymetrix, eBioscience, Thermofisher Scientific, Waltham, MA, USA; Cat#:562804); anti-human CD86 PE-Cyanine7, clone: IT2.2 (Affymetrix, eBioscience, Thermofisher Scientific, Waltham, MA, USA; Cat#: 25-0869-42); anti-human CD 80 PE, clone:2D10 (Affymetrix, eBioscience, Thermofisher Scientific, Waltham MA, USA; Cat#: 12-0809); anti-human CD274 (PD-L1) PerCP-eFluor 710, clone: MIH1 (Affymetrix, eBioscience, Thermofisher Scientific, Waltham, MA, USA; Cat#: 46-5983-42); anti-human CD3e FITC, clone: BW264/56 (Miltenyi Biotech, Auburn, CA, USA; Cat#:130-080-40); anti-human IL-17A APC-eFluor 780, clone: 64DEC17 (Affymetrix, eBioscience, Thermofisher Scientific, Waltham, MA, USA; Cat#47-7179-42); anti-human FOXP3 APC, clone 236A/E7 (Affymetrix, eBioscience, Thermofisher Scientific, Waltham, MA, USA; Cat#:17-477-72); anti-human CD25 PE, clone: 2A3 (STEMCELL Technologies, Vancouver, BC, Canada; Cat#: 10512); CTLA-4 PerCP-eFluor 710, clone: 14D3 (Affymetrix, eBioscience, Thermofisher Scientific, Waltham, MA, USA; Cat#:46-1529-42); PD1 Brilliant Violet 510, clone: EH12.1 (Affymetrix, eBioscience, Thermofisher Scientific, Waltham, MA, USA; Cat#:563076); and anti-human CD28 FITC, clone: CD28.2 (Affymetrix, eBioscience, Thermofisher Scientific, Waltham, MA, USA; Cat#: 11-0289-42).

### 4.8. Immune-Modulatory Effect of MoDC Exos on Recipient MoDCs

The immune-modulatory influence of MoDC exo subtypes on acceptor MoDCs was investigated by incubating 10^8^/mL MoDC esxo subtypes in MoDC culture medium on day 4. Cells were harvested on day 6. The expressions of activation markers, including HLADR, CD86, and CD80, as well as inhibitory marker PDL1, were analyzed through flow cytometry. mRNA expressions of inflammatory cytokines, including IL6, IL10, IL1B, TNFa, and IL23, were measured using qPCR.

### 4.9. T Cell Isolation, Activation, and Polariztion

Naïve T cells were isolated using an EasySep™ Human Naïve CD4+ T Cell Isolation Kit (Miltenyi Biotech, Auburn, CA, USA). T cell purity was assessed through a flow cytometry analysis of CD4 markers and were typically >95% pure. For direct T cell stimulation and proliferation, a 96-well u-bottom plate was coated with 10 µg/mL anti-CD3 antibody (Invitrogen, Thermofisher Scientific West Columbia, SC, USA; Cat#: 16-0037-85) and incubated at 37 °C with 5% CO2 for 2 h. The plate was then rinsed twice with PBS, and cells in the presence or absence of 10^8^/mL MoDC exo subtypes were added to the plate at 200,000 cells per well in complete RPMI 1640 medium (10% fetal bovine serum (FBS), 1% Pen Strep, 1x nonessential amino acids, 0.1% b-mercaptoetanol) containing 10 µg/mL anti-CD28 antibody (Invitrogen, Thermofisher Scientific West Columbia, SC, USA; Cat#:16-0289-85). Cells were incubated at 37 °C with 5% CO2 for 72 h, and T cell polarization/differentiation were assessed using flow cytometry. Regulatory T cells (Tregs) were identified by gating on double-positive cells for FOXP3 and CD25 in CD4^+^ cell gate populations, while T helper 17 induction was identified by measuring IL17A^+^ cells in CD4^+^ cell populations. Inhibitory molecules’ CTLA4 and PD1 expressions were measured by gating on CTLA4^+^ cells and PD1^+^ cells in CD4^+^ cell populations, respectively.

### 4.10. Exo Uptake and Confocal Microscopy

Dil-labeled exosomes were used to demonstrate the uptake of exosomes in vitro, as previously described [23]. Exosomes labeled with Dil (D282, Thermofisher Scientific, Waltham, MA, USA) were cocultured with MoDCs or CD4 T cells for 24 h. Cells were harvested, fixed with 4% paraformaldehyde, permeabilized with 0.1% Triton X-100, and stained on glass slides with phalloidin (A12379) and DAPI (D1306) (Invitrogen, Thermofisher scientific West Columbia, SC, USA). Images were captured with a Zeiss 780 upright confocal microscope (Carl Zeiss, AG, Oberkochen, Germany).

### 4.11. Statistical Analysis

Data analysis was performed using one-way ANOVA with a significance level at *p* < 0.05 and a 95% confidence interval, followed by Tukey’s or Bonferroni post hoc multiple-comparisons tests. The Shapiro–Wilk normality test was used to evaluate normal distribution assumption. When normality assumptions were not met, a nonparametric alternative test was used. Data are expressed as mean ± standard deviation (SD), and experiments were repeated 3 times. Data were analyzed using GraphPad Prism 9 (GraphPad Software 9.5.1, La Jolla, CA, USA).

## Figures and Tables

**Figure 1 ijms-24-11306-f001:**
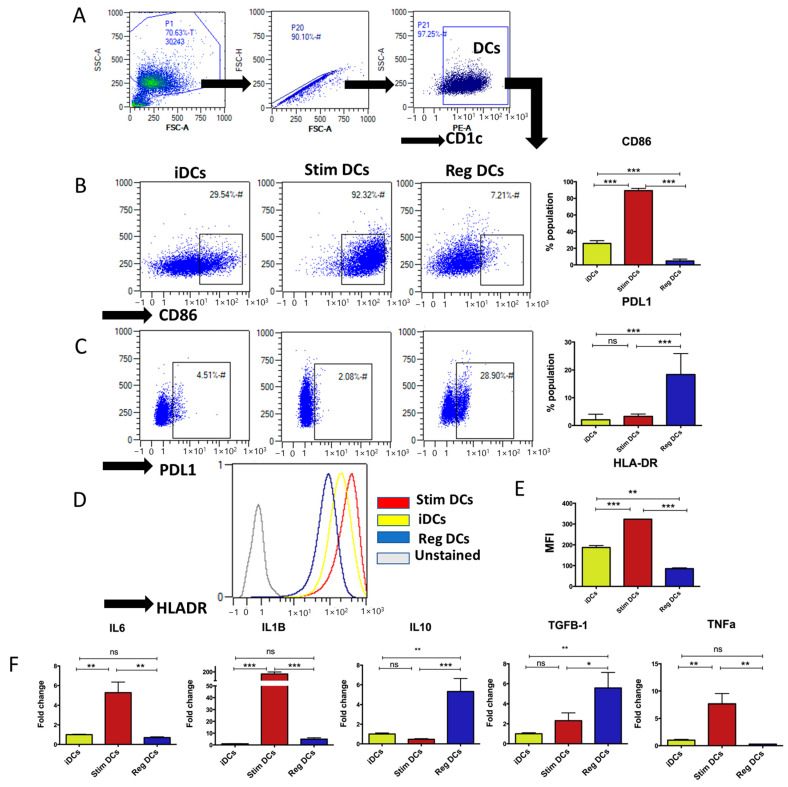
Characterization of donor MoDC subtypes. (**A**) Flow cytometry scattergrams showing CD1c^+^ cell (MoDC subtypes) gating strategy. (**B**) Flow cytometry scattergrams and summary bar graphs of population percentages of CD86^+^ cells and (**C**) PDL1^+^ cells in MoDC subtypes (iMoDCs, stimMoDCs, and regMoDCs). (**D**) Summary histograms and (**E**) bar graph of MFI of HLA-DR on MoDC subtypes. (**F**) Relative mRNA expressions of inflammatory cytokines (left to right) *IL-6, IL-1B, IL-10, TGF-B1,* and *TNFa* of different MoDC subtypes. ns = Not significant * *p* = 0.05, ** *p* = 0.01, and *** *p* = 0.001; one-way ANOVA and Tukey’s post hoc multiple comparison test.

**Figure 2 ijms-24-11306-f002:**
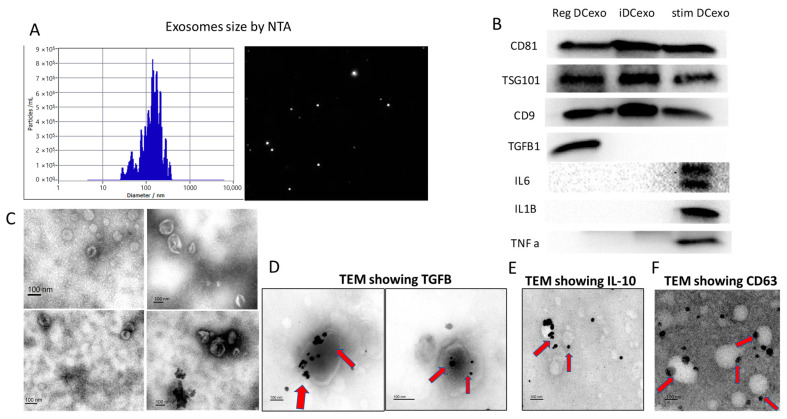
Characterization of human MoDC exo subtypes. (**A**) NTA revealed correct size range of MoDCexos of ~150 nm. (**B**) Immunoblot revealing shared and distinct proteins in MoDC exo subtypes, including exosomal markers C81, TSG101, and CD9; anti-inflammatory cytokine TGFB1; and proinflammatory cytokines IL6, IL1B, and TNFa. (**C**) SEM images showing cup-shaped MoDC exos; (**D**) immunogold TEM images of regMoDC exos revealing intraluminal and surface localization of (**D**) TGFB1 and (**E**) IL10, as well as (**F**) surface expression of exosomal marker CD63 (arrows).

**Figure 3 ijms-24-11306-f003:**
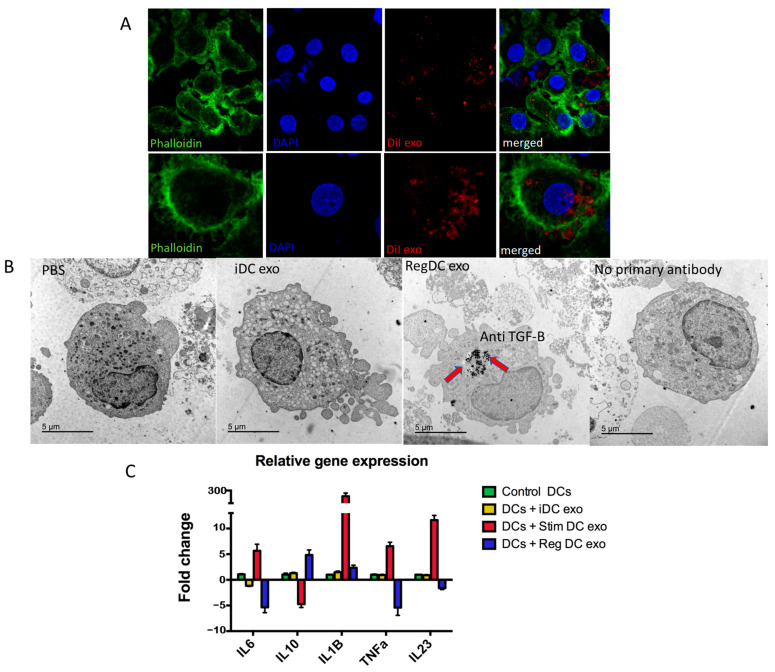
Recipient MoDCs take up exos, altering inflammatory cytokine profiles. (**A**) Dil (red)-labeled MoDC exos (arrows) inside recipient MoDCs counterstained with DAPI nuclear stain (blue), and phalloidin (green) for the cell membrane and visualized using confocal microscopy (upper panel images captured with 20× lens and lower panel with 63× lens). (**B**) Colocalization of TGFB1 in MoDCs +/− regMoDCexo or iMoDCexo treatment using immunogold TEM. (**C**) Q-rt-PCR through TaqMan gene expression using delta-delta CT and plotted as relative fold change of inflammatory cytokines *IL-6*, *IL-10*, *IL-1B*, *TNFa*, and *IL-23*.

**Figure 4 ijms-24-11306-f004:**
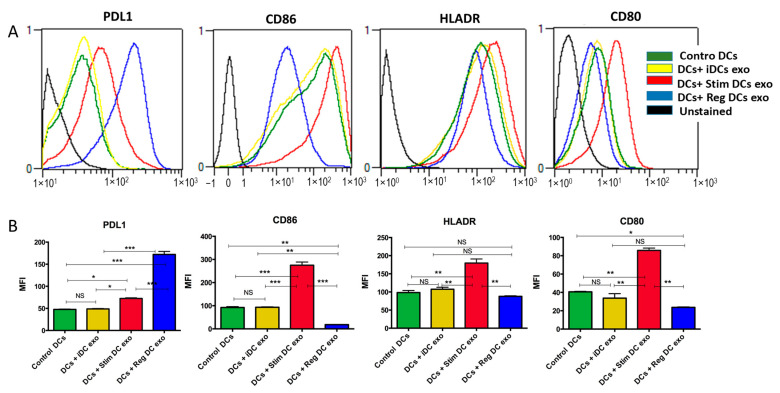
MoDCs exo subtypes modulate recipient MoDCs immunophenotype. (**A**) Flow cytometry histograms of PDL1-, CD86-, HLADR-, and CD80-positive MoDCs after coculturing with 10^8^/mL MoDC exo subtypes for 3 days. (**B**) Summary bar graphs of median fluorescent intensity (MFI) data for PDL1, CD86, HLADR, and CD80. NS = Not significant. * *p* = 0.05, ** *p* = 0.01, and *** *p* = 0.001; one-way ANOVA and Tukey’s post hoc multiple comparison test.

**Figure 5 ijms-24-11306-f005:**
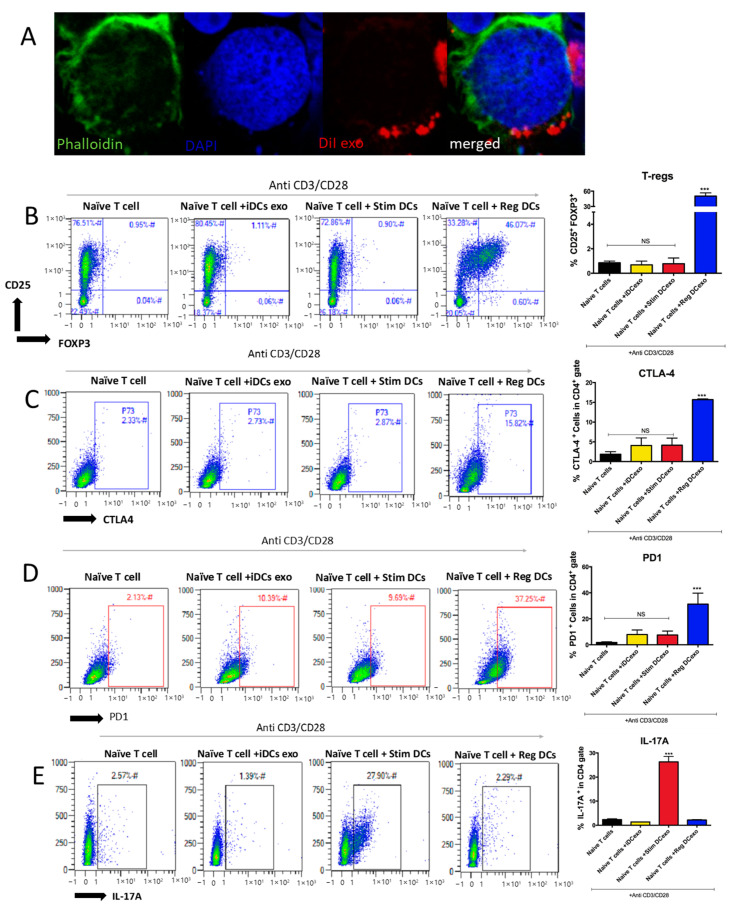
Modulating T cell effector responses by MoDCs exo subtypes. (**A**) Confocal microscopy images showing Dil (red)-labeled MoDC exos (arrows) colocalized with recipient CD4 T cells, counterstained with DAPI nuclear stain (blue) and phalloidin (green) for the cell membrane, and visualized using confocal microscopy. (**B**–**E**) FACS analysis and summary bar graphs of (from top to bottom) percentages of IL-17A, CD25+Foxp3+, CTLA4+, and PD1+ cells in naïve human CD4 T cells +/− anti-CD3/CD28 and +/− exos. NS = not significant. *** *p* = 0.001 determed with one-way ANOVA and Tukey’s multiple comparisons.

**Figure 6 ijms-24-11306-f006:**
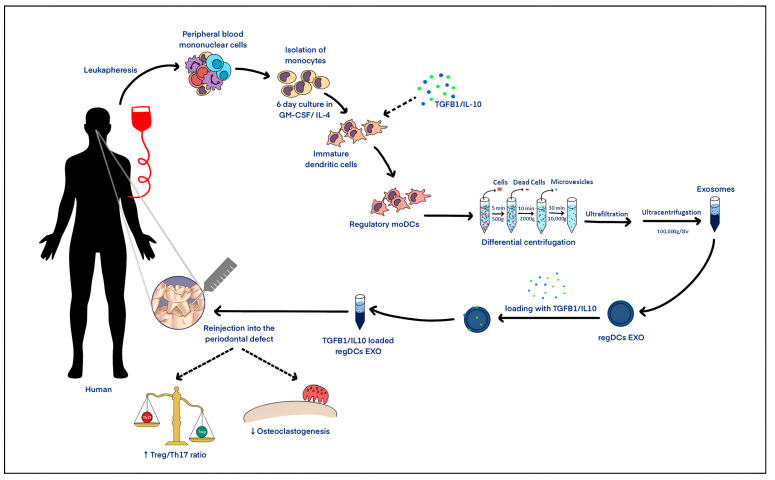
Proposed model for immunotherapeutic strategy to modulate host immune cells to treat periodontal disease.

## Data Availability

The raw data supporting the findings of this study will be made available by the authors upon reasonable request.

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
