# Peer review of "Engineered Human Dendritic Cell Exosomes as Effective Delivery System for Immune Modulation"

_ijms, 2023, doi:10.3390/ijms241411306_

Round 1

Reviewer 1 Report

The authors attempted to explain the significance of dendritic cell exosomes as an efficient delivery route for reprogramming a degenerative bone immune response in the article titled" Engineered Human Dendritic Cell Exosomes as Effective Delivery System for Reprogramming a Bone Degenerative Type Immune Response to a Bone Sparing Type Response." Although the authors put sufficient effort into designing the study and doing good experiments, they lack many specific points while writing the article. Here are some key points which are creating confusion while reading the article those are as follows:

1.       The article's title is very confusing. It would be better if the author could cut the title short of making it less confusing for the readers. Instead of writing about Bone Sparing and Bone Degenerative, authors can use periodontitis.

2.       The authors used the term Engineered Human Dendritic Cell Exosomes but have not mentioned anywhere in the article what and how these exosomes are engineered.

3.       The introduction lacks specific information about the Delivery System, Reprogramming of the immune system, and Bone Sparing Type Response, which is the article's title.

4.       The authors mentioned that exosomes are nano-sized membrane-enclosed particles but did not mention the size of these particles.

5.       The article does not mention exactly how many cells are used to extract the exosomes. The extracted exosome particle number per cell is also not mentioned anywhere.

6.     How do these exosomes are prepackaged with the immunoregulatory cargo?

7.       Figure legends are not giving enough information about the figure. All the figure legends should be elaborative and explain all the parameters used, including types of antibodies, tagging, plotting strategy, and experiment idea. Authors can follow the following article for reference: Exp Mol Med 55, 665–679 (2023). https://doi.org/10.1038/s12276-023-00949-7

8.       Figure 1 A is a bit confusing and unclear; it is not well explained in the text.

9.       Figure 3 A is not well-focused. It would be good if it could be generated again with good resolution. In the colocalization experiment, the merged image is unclear. It would be good if the authors could change the colors used for phalloidin (magenta)  and the exosome (red) to green.

10.   The first paragraph of the discussion can go to the introduction.

11.   It Would be good if the authors could use Z-Stack Image to show the colocalization.

12.   Material methods segments are not clear on how the experiments were performed. How many monocytes were used to generate Dendritic cells? The ratio of exosomes per number of cells has not been disclosed. Overall the material method section needs to be rewritten with detailed explanations.

Author Response

We thank the reviewers for the invaluable comments. We greatly appreciate your valuable feedback regarding our manuscript. All suggestions have been carefully considered and addressed as follow:

Reviewer 1:

The authors attempted to explain the significance of dendritic cell exosomes as an efficient delivery route for reprogramming a degenerative bone immune response in the article titled" Engineered Human Dendritic Cell Exosomes as Effective Delivery System for Reprogramming a Bone Degenerative Type Immune Response to a Bone Sparing Type Response." Although the authors put sufficient effort into designing the study and doing good experiments, they lack many specific points while writing the article. Here are some key points which are creating confusion while reading the article those are as follows:

  1. The article's title is very confusing. It would be better if the author could cut the title short of making it less confusing for the readers. Instead of writing about Bone Sparing and Bone Degenerative, authors can use periodontitis.
  • This has been addressed in the updated title.
  1. The authors used the term Engineered Human Dendritic Cell Exosomes but have not mentioned anywhere in the article what and how these exosomes are engineered.
  • The engineering strategy has been now clarified in the revised manuscript at lines 55-57 (introduction section), 195-198 (materials and methods section).
  1. The introduction lacks specific information about the Delivery System, reprogramming of the immune system, and Bone Sparing Type Response, which is the article's title.
  • This has been now addressed in the revised version. Based on the reviewers’ suggestions, the term ‘’reprograming’’ is now replaced with ‘’modulating” throughout the revised manuscript with focus on the DCs and T cells responses and their relevance to bone resorbing cells (osteoclast) activation. This has been stated in the updated manuscript at lines 31-44. and the development of the delivery system is mentioned in lines 55-57.
  1. The authors mentioned that exosomes are nano-sized membrane-enclosed particles but did not mention the size of these particles.
  • This is mentioned in lines 45, 79, 86 and

  1. The article does not mention exactly how many cells are used to extract the exosomes. The extracted exosome particle number per cell is also not mentioned anywhere.
  • This has been now detailed in the methodology section at lines 177-183 and 201-207.
  1. How do these exosomes are prepackaged with the immunoregulatory cargo?
  • This has been now detailed in the methodology section at lines 195-198 (materials and methods section).
  1. Figure legends are not giving enough information about the figure. All the figure legends should be elaborative and explain all the parameters used, including types of antibodies, tagging, plotting strategy, and experiment idea. Authors can follow the following article for reference: Exp Mol Med 55, 665–679 (2023). https://doi.org/10.1038/s12276-023-00949-7
  • This has now been addressed in the revised version. Kindly find the revised document with track changes.
  1. Figure 1 A is a bit confusing and unclear; it is not well explained in the text.
  • This has now been addressed in the revised version. The figure has been updated for clarification. Kindly find the revised document with track changes.
  1. Figure 3 A is not well-focused. It would be good if it could be generated again with good resolution. In the colocalization experiment, the merged image is unclear. It would be good if the authors could change the colors used for phalloidin (magenta) and the exosome (red) to green.
  • This has now been addressed in the revised version. Kindly find the revised document with track changes. Figure is now updated.
  1. The first paragraph of the discussion can go to the introduction.
  • This has been done in the revised version and the paragraph is now moved to the introduction section at lines 30-42. Kindly find the revised document with track changes.
  1. It Would be good if the authors could use Z-Stack Image to show the colocalization.
  • Thank you for the great suggestions. We now performed additional experiment using immunogold TEM to show colocalization of TGFB1 in MoDCs treated with regMoDCs exo, suggesting uptake/ interaction of exo with recipient cells and delivery of the therapeutic cargo (Fig 3B).

  1. Material methods segments are not clear on how the experiments were performed. How many monocytes were used to generate Dendritic cells? The ratio of exosomes per number of cells has not been disclosed. Overall the material method section needs to be rewritten with detailed explanations.
  • Thank you for the comments. The number of Monocytes and DCs used and the ratio of exo per cell has been now detailed in the methodology section at lines 177-183 and 201-207. The material and method section has been now revised and modified with more details on each experiment. Kindly find the revised document with track changes.

Reviewer 2:

The manuscript by Cutler et al, entitled “Engineered Human Dendritic Cell Exosomes as Effective Delivery System for Reprogramming a Bone Degenerative Type Immune Response to a Bone Sparing Type Response”, with the manuscript ID ijms-2334285, submitted to IJMS addresses the use of human dendritic cells-derived exosomes to modulate the functionality of human dendritic cells and T cells, in vitro. Despite the relevant presentation, some issues need to be revised by the reviewers in order to improve the quality of the manuscript.

1-As the research team has previously published a very relevant manuscript on the topic, with a murine model, but further disclosing a thorough in vitro and in vivo characterization, the relevance of the present study should be further highlighted. 

  • The relevance and rationale has been now highlighted at lines 52-55, 57-60, 130-134.

2-The contextualization within periodontal disease and potential influence on bone sparing modulation is excessively speculative. Authors only address the in vitro cell response of human DCs and T-cells and the major focus of the manuscript should be limited to this interaction. As no pathological condition or model was used, authors should refrain from extrapolating into PDs.

We thank the reviewer for this comment. In response, we have carefully revised the focus of our research to emphasize the invitro modulatory effect of human MoDCexo on key immune cells involved in inflammatory diseases such as PD. This updated manuscript includes modifications to the title, introduction, and discussion sections, which now highlight the significant immune modulatory effect of MoDCexo in human DCs and T cells in vitro and the potential implications of this exosome based nano-delivery system for the treatment of PD.

3- The title is also excessively speculative and should only focus on the observed interactions.

  • This has been addressed in the revised version. Kindly see the modified title.

4-As no assay was conducted to address the modulation of the bone response, theis speculative information should be reduced to a minimal supposition. 

  • We agree with the reviewer. This has been addressed in the revised version. Kindly find the revised document with track changes.

5-Introduction should address the relevance of the evaluated cell populations in physiological and pathological conditions, DC exos, typical exos-cargo, and therapeutic potential of the approach. The contextualization on PD is excessive and out of focus.

  • This has been addressed in the introduction at lines 31-60 with focus on the DCs and T cells immune modulation and their relevance to bone resorbing cells (osteoclast) activation.

6-The same for the Discussion section that needs a significant revision, more focused on the biological outcomes and relevance of the attained findings on cell modulation.

  • This has been addressed in the revised version. Kindly find the revised document with track changes.

7-The use of the word “reprograming” is excessive as there is no evidence of a sustained effect in the absence of the delivery. Authors should use “modulation” instead. 

  • The term ‘’reprograming’’ is now replaced with ‘’modulating’’ throughout the manuscript with focus on the DCs and T cells responses. Kindly find the revised document with track changes.

8- Confocal images should be complemented in order to show that exos are inside the cells and not only over the cells.

  • Thank you for the great suggestions. We now performed additional experiment using immunogold TEM to show colocalization of TGFB1 in MoDCs treated with regMoDCs exo, suggesting uptake/ interaction of exo with recipient cells and delivery of the therapeutic cargo (Fig 3B).

9-TEM imaging should also be used to show internalization and cynetic profile of the exos.

  • TEM has been used to exo profile (Fig 2 C, D and E). An additional experiment was performed using immunogold TEM to show colocalization of TGFB1 in MoDCs treated with regMoDCs exo, suggesting uptake/ interaction of exo with recipient cells and delivery of the therapeutic cargo (Fig 3B).

Reviewer 2 Report

The manuscript by Cutler et al, entitled “Engineered Human Dendritic Cell Exosomes as Effective Delivery System for Reprogramming a Bone Degenerative Type Immune Response to a Bone Sparing Type Response”, with the manuscript ID ijms-2334285, submitted to IJMS addresses the use of human dendritic cells-derived exosomes to modulate the functionality of human dendritic cells and T cells, in vitro. Despite the relevant presentation, some issues need to be revised by the reviewers in order to improve the quality of the manuscript.

-        As the research team has previously published a very relevant manuscript on the topic, with a murine model, but further disclosing a thorough in citro and in vivo characterization, the relevance of the present study should be further highlithed.

-        The contextualziation within periodontal disease and potential influence on bone sparing modulation is excessively speculative. Authors only address the in vitro cell response of human DC and T cells and the major focus of the manuscript should be limited to this interaction. As no pathological condition or model was used, authors should refrain from extrapulating into PDs.

-        The title is also excessively speculative and should only focus on the observed interactions.

-        As no assay was conducted to address the modulation of the bone response, theis speculative information should be reduced to a minimal supposition.

-        Introduction should address the relevance of the evaluated cell populations in physiological and pathological conditions, DC exos, typical exos-cargo, and therapeutic potential of the approach. The contextualization on PD is excessive and out of focus.

-        The same for the Discussion section that needs a significant revision, more focused on the biological outcomes and relevance of the attained findings on cell modulation.

-        The use of the word “reprograming” is excessive as there is no evidence of a sustained effect in the absence of the delivery. Authors should use “modulation” instead.

-        Confocal images should be complemented in order to show that exos are inside the cells and not only over the cells.

TEM imaging should also be used to show internalization and cynetic profile of the exos.  

Round 2

Reviewer 1 Report

The authors have made significant changes in the article "Engineered Human Dendritic Cell Exosomes as Effective Delivery System for Immune Modulation." Although looking at the quality of the article, images, and detailed descriptions, the manuscript now has a higher chance of getting published. There are a few minor changes or clarifications required those are as follows:

The authors have proposed a model for an immunotherapeutic strategy to modulate host immune cells to treat periodontal diseases. The abstract does not mention it, but the introduction and discussion are focused on periodontal diseases, which makes the article a bit confusing. I will suggest authors write a few lines in the discussion and introduction about the potential use instead of focusing on periodontal diseases in the introduction and discussion.

Another suggestion would be to focus the article's theme on "Dendritic Cell Exosomes as an Effective Delivery System" instead of periodontal diseases because no data is provided regarding the condition.

The authors have attached duplicate figures with different coloring patterns for most images. Delete the repetitive figure and content to make it clear for the readers.

Author Response

I would like to thank the reviewer for his comments.

The PI's work has focused on the immunopathogenesis of periodontitis for several decades by conducting immunohistochemical human studies. Based on these results  we have now identified a potential novel immunotherapeutic approach to this disease, first by conducting mouse research, both in vitro and in vivo, then human studies.  The current article follows the logical progression of this work by using  custom exosomes isolated from human cells and showing that they  are similarly effective in modulating the DC and T cell responses in vitro , consistent with what we have published in the mouse. Thus we cannot ignore the relevance to periodontitis. We have further emphasized this in the newly revised abstract (please see track changes)to be cohesive with the introduction and discussion. Since the article contains no data from the human disease  per se we have removed that from the title (ie.. theme).  The repetitive figures have been deleted even though they still appear in the track changes. A revised PDF without track changes has been uploaded.  

Reviewer 2 Report

The reviewer acknowledges the authors' effort to improve the quality of the manuscript. 

Author Response

We would like to thank the reviewer for his comments.